# Oscillating modes of driven colloids in overdamped systems

Johannes Berner [1,2], Boris Müller[3,4], Juan Ruben Gomez-Solano[1,2], Matthias Krüger[3,4,5] &
Clemens Bechinger[1,2,4]

Microscopic colloidal particles suspended in liquids are a prominent example of an over-damped system where viscous forces dominate over inertial effects. Frequently, colloids are used as sensitive probes, e.g., in biophysical applications from which molecular forces are inferred. The interpretation of such experiments rests on the assumption that, even when the particles are driven, the liquid remains in equilibrium. Here we experimentally demonstrate that this is not valid for particles in viscoelastic fluids. Even at small driving forces, we observe particle oscillations with several tens of seconds. They are attributed to non-equilibrium fluctuations of the fluid being excited by the particle's motion. The oscillatory dynamics is in quantitative agreement with an overdamped Langevin equation with negative friction-memory term being equivalent to a stochastically driven underdamped oscillator. Such oscillatory modes are expected to widen the use of colloids as model systems but must also be considered in colloidal probe experiments.

[1] 2. Physikalisches Institut, Universität Stuttgart, D-70569 Stuttgart, Germany. [2] Fachbereich Physik, Universität Konstanz, D-78464 Konstanz, Germany. [3] Institut für Theoretische Physik IV, Universität Stuttgart, D-70569 Stuttgart, Germany. [4] Max-Planck-Institut für Intelligente Systeme, D-70569 Stuttgart, Germany. [5] Present address: Universität Göttingen, Institut für Theoretische Physik, D-37077 Göttingen, Germany. Correspondence and requests for materials should be addressed to C.B. (email: clemens.bechinger@uni-konstanz.de)

Brownian motion is a paradigmatic example of a Markovian process where each incremental step along the particle's trajectory is fully determined by its previous position[1,2]. Such memory-free behaviour is valid for time scales larger than the relaxation times of the fluid and inertial relaxation times of the colloid (typically below nanoseconds) where the collisions with the solvent's molecules can be regarded as an entirely random process[2]. As the velocity distribution of the molecules, i.e., that of the thermal bath, is not influenced by the colloidal's motion, it can be considered as a true, inert thermostat, providing purely white noise. Many experiments confirmed, that this assumption remains valid even when the colloid is subjected to external driving forces (see, e.g., Ref. [3] and references therein). Accordingly, the postulation of a weak coupling of colloidal particles to the thermal bath, as, e.g., considered within the framework of stochastic thermodynamics[4,5], provides a faithful description of the non-equilibrium properties of such systems.

The assumption of a rapidly relaxing thermal bath is not applicable to viscoelastic fluids like semi-dilute polymer solutions, micellar systems, or dense colloidal suspensions. Such systems are characterised by stress relaxation times $\tau_s$ comparable or even larger than that of the colloidal motion[2,6]. Accordingly, when colloidal particles are driven through such a fluid (e.g., by means of an optical trap), it can not be regarded to remain in equilibrium. Theoretical studies predicted that in this regime the particle dynamics becomes largely affected by the fluid's non-equilibrium microstructural deformations, and that the measured viscosity may exhibit a non-trivial dependence on the trap stiffness[7]. In particular for large driving velocities (high shear rates), several experiments reported the occurrence of unsteady particle motion[8,9] and strong deviations from the behaviour in simple viscous liquids[10–19]. These findings originate from the nonlinear rheological properties in viscoelastic fluids (e.g., shear thinning), which is generally observed in micro- and macro-rheological experiments[2,11,17,20].

In contrast, the experiments presented here are performed at low driving velocities, where the viscosity is constant, and within the linear response regime. When we analyse the motion of the particle inside the harmonic optical trap, which moves with constant velocity through a worm-like micellar solution, we observe a new harmonic oscillator state with non-trivial fluctuations. Despite all motion being overdamped, it shows oscillating (underdamped) modes, which are strictly ruled out in equilibrium systems. These oscillations are accompanied by large fluctuation amplitudes, so that the particle's mean squared displacement (MSD) is drastically different from the equilibrium one. Although the main text focuses on a worm-like micellar solution, we observe similar particle oscillations in other viscoelastic fluids comprising different chemistry and microstructure (see Methods). Therefore, we believe that the reported oscillations are a generic feature of particles in non-equilibrium baths.

## Results

**Experiments**. Our experiments are performed in an equimolar solution of surfactant, cetylpyridinium chloride monohydrate (CPyCl) and sodium salicylate (NaSal) in deionised water at a concentration of 7 mM and at temperature $T = 298 \pm 0.2$ K. Under such conditions, these mixtures form an entangled viscoelastic network of worm-like micelles[21] with a structural relaxation time $\tau_s = 2.5 \pm 0.2$ s determined by a recoil experiment[19], where the length of the worm-like micelles is typically found in between 100 and 1000 nm[22], and the typical mesh size is on the order of 30 nm[23]. A small amount of silica particles with

diameter $2R = 2.73$ μm is added to this fluid and a single particle is optically trapped by a focused laser beam, which creates a parabolic potential $\frac{1}{2}\kappa\xi^2$ (with $\xi$ a spatial coordinate relative to the potential minimum) whose stiffness $\kappa$ is fixed by the laser intensity (Fig. 1a, b). The trap position, which is adjusted by a computer-controlled mirror, performs a one-dimensional motion with constant velocity $v$. The particle motion $\xi(t)$ relative to the trap centre is measured with a rate of 145 fps (for further details, see Methods). From the mean position $\langle\xi\rangle$ and by applying Stokes' law, we obtain the fluid's (micro-)viscosity $\eta \equiv \kappa|\langle\xi\rangle|/6\pi v R$. Figure 1c demonstrates that—similar to a Newtonian liquid—$\eta$ is independent of $v$ for the parameters used in this study, and that our experiments are performed within the linear response regime. In that range, the dimensionless Weissenberg number $\text{Wi} = v\tau_s/2R$, is well below one. $v/2R$ estimates the shear rate near the driven particle, and for the given values between 0.02 and 0.1 s$^{-1}$, bulk rheological measurements[9] indeed find the zero-shear viscosity. The Reynolds number is of order $10^{-9}$, so that inertial effects in the fluid are negligible.

Figure 1d and e compare the particle motion in equilibrium (Wi = 0) and for non-equilibrium conditions (Wi = 0.34). In the following, particle fluctuations around its mean position are quantified by $x(t) = \xi(t) - \langle\xi(t)\rangle$. As expected, in equilibrium the particle performs random fluctuations within the trap and the distribution of $x(t)$ is in excellent agreement with Boltzmann statistics. In contrast, considerable deviations from the equilibrium probability distribution are observed at finite (but very small) Wi. Such unexpected behaviour is supported by the corresponding MSD, $\langle(x(t) - x(0))^2\rangle$, which are shown in Fig. 2 for four different Wi. In equilibrium, the MSD grows monotonically and saturates at $2k_BT/\kappa$, in accordance with the equipartition theorem. For finite Wi, however, the MSDs grow considerably above this value. The particle explores a larger configurational space within a moving (compared with a static) trap.

The trajectories in Fig. 1d, e reveal another, even more striking difference between equilibrium and non-equilibrium: in contrast to the random particle fluctuations in equilibrium, the data in Fig. 1e are qualitatively different and appear to exhibit oscillatory particle motion. To analyse such unexpected behaviour in more detail, we study the conditional probability $P(x, t|x_0, 0)$ to find a particle at position $x$ at time $t$, given that it was at $x_0$ at $t = 0$. Accordingly, such mean conditional displacements (MCDs) are given by $\langle x\rangle_{x_0}(t) \equiv \int dx P(x, t|x_0, 0)x$. Experimentally, MCDs with different initial positions $x_0$ are obtained from (long) trajectories by using any (random) occurrence $x(t) = x_0$ as an initial point. We have verified, that such curves scale linearly in $x_0$ (Methods).

In equilibrium, the MCD decays monotonically on a time scale roughly given by the ratio of the particle's friction and the trap stiffness $\kappa$ (Fig. 3, Wi = 0)[24]. Such monotonic behaviour is expected for any complex fluid, because the Fokker–Planck operator, including colloid and the surrounding micelles, has real negative eigenvalues. Therefore, the MCD is a sum of positive exponentially decaying functions[2,24] (Methods). A qualitatively different behaviour, however, is observed in the non-equilibrium steady state: here, the MCDs do not decay monotonically, but show oscillations whose amplitudes increase with Wi (Fig. 3). The oscillation time decreases with increasing Wi and is for Wi = 0.04 —the slowest drive accessible in our experiments—about 100 s, so that this curve relaxes much slower than the equilibrium curve (compare the time axes).

Figure 4 shows the dependence of amplitude and frequency of oscillations on Weissenberg number, where for both quantities, a gradual decrease towards equilibrium (Wi = 0) is observed.

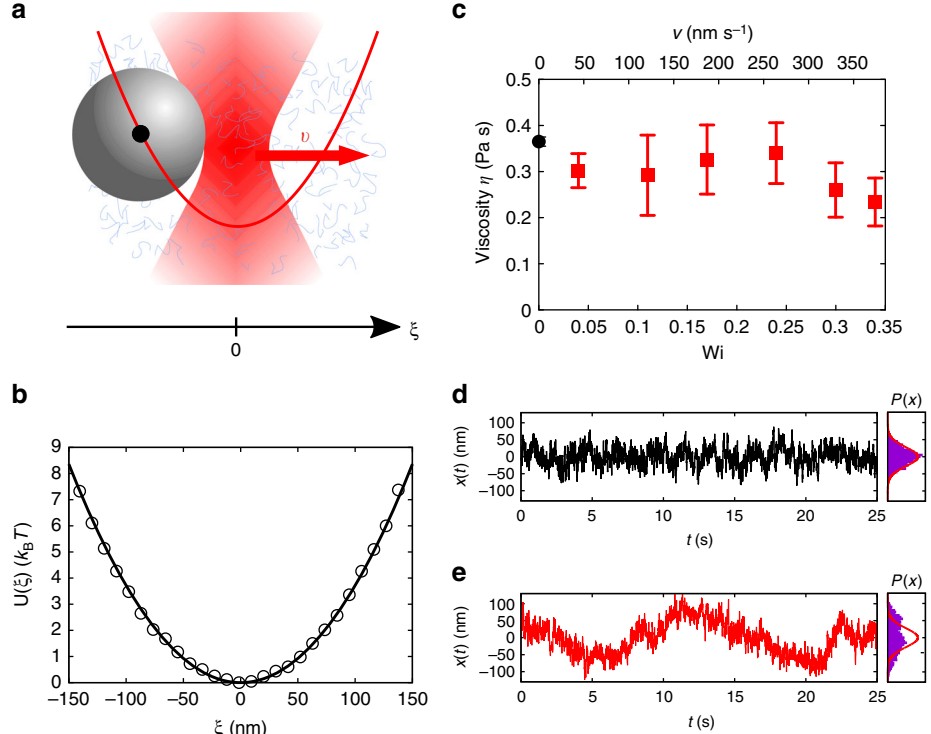

**Fig. 1** Colloidal particle in a harmonic trap driven through a viscoelastic fluid. **a** Micrometre-sized particle in a micellar solution and subjected to a harmonic potential, which moves with constant velocity $v$. **b** Measured trap potential (symbols) and a parabolic fit (solid line), from which the trap stiffness $\kappa$ is extracted. **c** Viscosity $\eta$ obtained from the drag force acting on the particle as a function of the trap velocity and the corresponding Weissenberg number, respectively. The value at Wi = 0 is obtained from the mean squared displacement of the particle in the absence of the trap. $\eta$ is independent of the particle velocity, indicating the linear response regime. In contrast to Newtonian liquids, we observe strong differences in the fluctuations and their probability distributions $P(x)$, i.e., for Wi = 0 (**d**) and for finite Wi (**e**). Red lines give the Boltzmann distribution in equilibrium

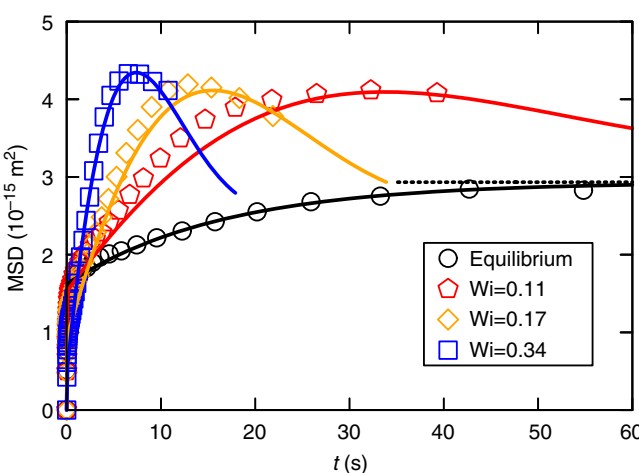

**Fig. 2** Mean squared displacements. MSDs are monotonic in equilibrium and saturate to $2k_BT/\kappa$ (horizontal dashed line), while they grow much quicker and higher for finite driving. It is noteworthy that the shown Weissenberg numbers are in the linear response regime (see Fig. 1c). Lines show theoretical results introduced below

**Theory.** To understand the origin of particle oscillations in an overdamped system, we recall that on time scales beyond microseconds, the particle's motion results from the balance of a frictional force, stochastic 'noise' and the optical force[2]. For the micellar bath, the mean frictional force at time $t$ is a nonlinear functional $\mathcal{G}\{\dot{x}(t') + v\}_{t' \leq t}$ of the past trajectory, and so is the noise (see, e.g., Ref. [25] and see Refs. [26–29] for other approaches).

Encouraged by the observation that the experimental MCDs are linear in $x_0$ (Methods), we proceed by considering a linear equation for $x$,

$$\int_{-\infty}^{t} ds\, \dot{x}(s)\, \Gamma^{(v)}(t-s) = -\kappa x(t) + f^{(v)}(t). \quad (1)$$

Formally, $\Gamma^{(v)}$ is the functional derivative of $\mathcal{G}$ around the non-equilibrium steady state,

$$\Gamma^{(v)}(t-s) = \left.\frac{\delta \mathcal{G}\{\dot{x} + v\}(t)}{\delta \dot{x}(s)}\right|_{\dot{x}=0}, \quad (2)$$

and similarly for the noise $f^{(v)}$, which is then independent of $\dot{x}$, and $\langle f \rangle = 0$. The nonlinearity of $\mathcal{G}$ makes the transformation to the co-moving frame as well as the linearisation non-trivial, so that $\Gamma^{(v)}$ depends on $v$ (and also on $\kappa$[30]). $\Gamma^{(v)}(\tau) = 0$ for $\tau < 0$ (causality).

From Eq. (1), the MCDs are readily obtained by taking the mean with initial condition $x(t=0) \equiv x_0$ (it is noteworthy that velocities average to zero for $t < 0$),

$$\langle \hat{x} \rangle_{x_0}(s) = \frac{x_0 \hat{\Gamma}^{(v)}(s)}{s\hat{\Gamma}^{(v)}(s) + \kappa}, \quad (3)$$

with Laplace transforms $\hat{h}(s) = \int_0^\infty dt\, e^{-st} h(t)$. Notably, Eq. (3) is independent of noise $f^{(v)}$ and the MCDs are uniquely related to the memory kernel $\Gamma^{(v)}$.

The equilibrium curve in Fig. 3 can already be understood qualitatively from a simple model by Maxwell[31] or by

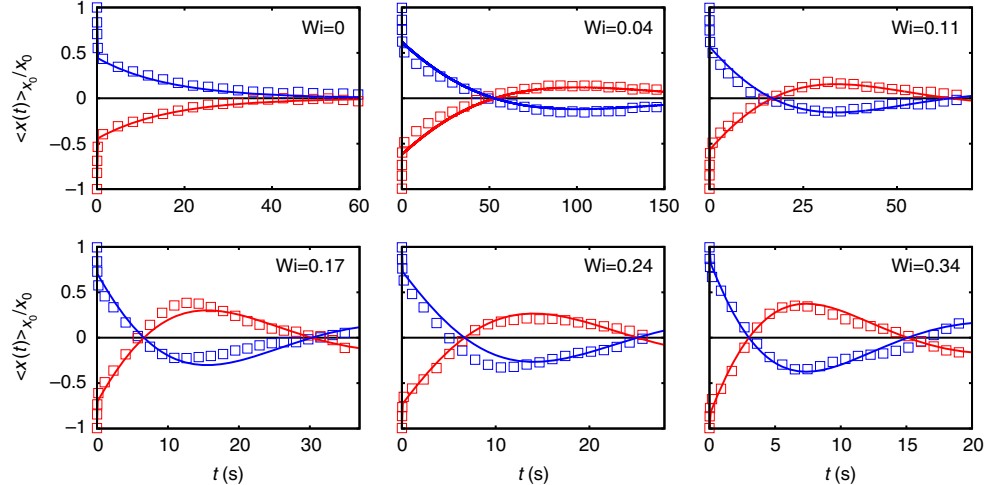

**Fig. 3** Oscillating modes. In equilibrium, the MCD relaxes exponentially, as expected for any complex fluid. For finite driving, MCDs show pronounced oscillations, which, especially for small Wi, drastically increase the system's correlation time (e.g., more than 150 s for Wi = 0.04). This behaviour is captured in a simple theoretical model (lines)

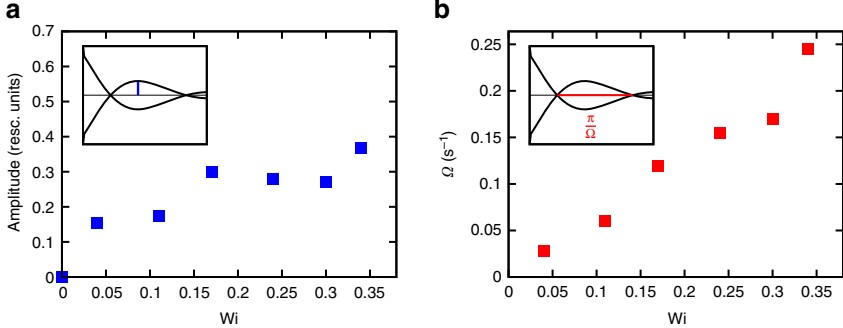

**Fig. 4** Oscillation amplitude and frequency. Oscillation amplitude (**a**) and frequency (**b**) of the MCD curves (shown in Fig. 3) vs. Wi. Insets illustrate how these quantities are derived from the experimental data. Both curves decrease with decreasing Weissenberg number, i.e., towards equilibrium. Regarding frequency, this implies that oscillations are particularly slow at small Wi. As the second root is not visible for Wi = 0.04 in Fig. 3, the corresponding value of $\Omega$ has been obtained from the time difference between the extremum and the first root

Jeffreys[32,33], which considers a memory time $\tau$,

$$\Gamma^{(0)}(t) = 2\gamma_\infty \delta(t) + \frac{\gamma_0 - \gamma_\infty}{\tau} e^{-\frac{t}{\tau}}. \qquad (4)$$

Here, $\gamma_\infty$ and $\gamma_0$ are friction coefficients at infinite and zero frequencies, respectively. We adjust the friction coefficients and relaxation time of this model in such manner to obtain best agreement with the experimental data. The result is shown as lines in Fig. 3, for parameters we refer to the Methods section. As expected, the MCD decays monotonically to zero in accordance with the experimental curve.

Aiming at a simple model for the non-equilibrium oscillations, we amend Eq. (4) by another generic exponential term to account for finite driving,

$$\Gamma^{(v)}(t) = 2\gamma_\infty \delta(t) + \frac{\gamma_0 - \gamma_\infty - \gamma_1}{\tau} e^{-\frac{t}{\tau}} + \frac{\gamma_1}{\tau_1} e^{-\frac{t}{\tau_1}}, \qquad (5)$$

and we use $\tau_1 > \tau$ throughout. The parameters in Eq. (5) may depend on Weissenberg number. Importantly, the new coefficient $\gamma_1$ is negative, so that $\Gamma^{(v)}(t)$ is negative for long times in contrast to the equilibrium kernel (Methods). Negative memory is a concept, which has been used in other fields of rheology of complex systems, e.g., when applying macroscopic shear. If such

shear is started abruptly, one sometimes observes so called stress overshoots, where the stress goes through a maximum as a function of time, once the yield stress is overcome[20,34–37]. Theoretically, these overshoots have been described by negative memory, as found from microscopic derivations in Refs.[20,37]. Again, we emphasise that, in equilibrium, $\Gamma$ is related to the force autocorrelation function (Methods), which is strictly positive on overdamped time scales.

When fitting the form of Eq. (5) for best agreement with experiments, there is one value which we preset: $\gamma_0$ may be identified with the viscosity shown in Fig. 1c, so that it is not varied in the fitting. (As detailed in the Methods section, for the larger driving, above Wi = 0.11, we used one exponential term additional to Eq. (5) to obtain quantitative agreement).

The MCDs so obtained are shown in Fig. 3 as solid lines, which reproduce well the experimental observations. After an initial decay, the curves oscillate as a function of time. Notable, the final relaxation can be on time scales that are much larger than both $\tau$ and $\tau_1$, so that the long relaxation times observed experimentally (e.g., for Wi = 0.04) are also found from Eq. (5). Conceptually different to the above mentioned studies on macroscopic shear, it is the additional presence of the optical trap, which, in

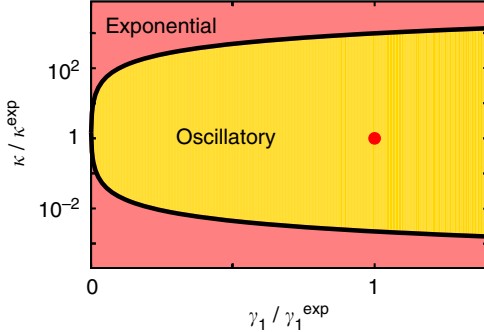

**Fig. 5** Dynamical phase diagram. The model of Eq. (5) yields distinct solutions, either decaying exponentially or showing oscillations, here shown for Wi = 0.04 as a function of $\kappa$ and $\gamma_1$ normalised on the corresponding experimental parameters $\kappa^{\text{exp}} = 2.3\,\mu\text{Nm}^{-1}$ and $\gamma_1^{\text{exp}} = -1135.1\,\mu\text{Nsm}^{-1}$, i.e., those obtained from the fit curve in Fig. 3

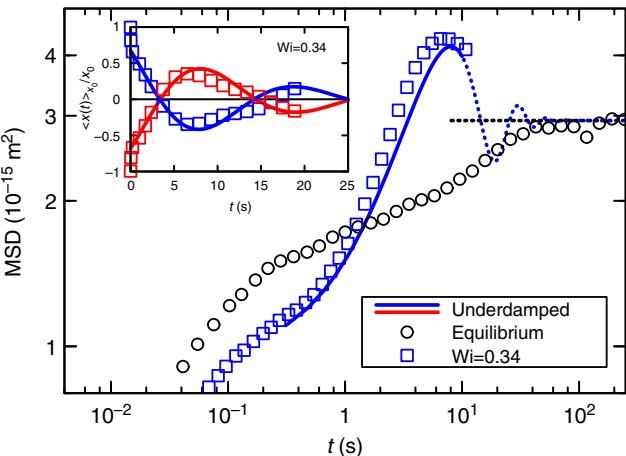

**Fig. 6** Effective mass. Comparison of experimental results and the solution for a massive colloid for Wi = 0.34. Main graph shows the MSD (compare Fig. 2, note the log scale), whereas the inset shows the corresponding MCD (compare Fig. 3) for the same Wi

combination with the negative memory, yields oscillatory solutions in our experiments.

It is noteworthy that the MCDs from Eq. (5) show two distinct types of solution: depending on the parameter values, there are either purely exponential solutions, $\sim e^{-|\nu|t}$ or damped oscillating modes, $\sim e^{-(|\nu|+i|\Omega|)t}$, the latter characterised by a finite $\Omega$ value.

Figure 5 shows the phase diagram as a function of $\kappa$ and $\gamma_1$. Notably, oscillations appear only for a finite range of $\kappa$ (with all other parameters fixed). Exceeding this range yields a purely exponential decay. In other words, for a given viscoelastic fluid with a well-defined relaxation time, oscillations occur only within a narrow range of stiffness of the optical trap, where conditions are resonant. This might explain why such oscillations have not been observed in previous experiments where a colloidal particle was dragged with an optical tweezer through a viscoelastic colloidal suspension[12]. In order to maximise the oscillatory behaviour in our experiments, the value of $\kappa$ was chosen to be in the centre of the oscillating phase. Indeed, a variation of the trap stiffness to smaller and larger values yields a much less pronounced oscillatory behaviour (Methods). The phase boundary shown in Fig. 5 can be analytically determined to

$$\kappa_c = \frac{\gamma_0 - 2\gamma_1}{\tau_1 - \tau} \pm \frac{2\sqrt{\gamma_1(\gamma_1 - \gamma_0)}}{\tau_1 - \tau}, \quad (6)$$

showing a critical point at $|\gamma_1| = 0$, from which $\kappa_c$ grows as a square root in $|\gamma_1|$.

From Eq. (1), we also obtain the corresponding MSDs

$$\langle(x(t) - x(0))^2\rangle = \frac{1}{\pi}\int_{-\infty}^{\infty}\mathrm{d}\omega\left(1 - e^{i\omega t}\right)\frac{\langle|\tilde{f}^{(\nu)}(\omega)|^2\rangle}{|\omega\tilde{\Gamma}^{(\nu)}(\omega) - i\kappa|^2}, \quad (7)$$

with Fourier transforms $\tilde{h}(\omega) = \int_{-\infty}^{\infty}\mathrm{d}t\,e^{-i\omega t}h(t)$. In contrast to the mean conditional curves, the MSDs involve the noise correlation, which in equilibrium is determined by the fluctuation-dissipation theorem (FDT)[38]

$$\left\langle|\tilde{f}^{(0)}(\omega)|^2\right\rangle = 2k_\mathrm{B}T\Re\left[\tilde{\Gamma}^{(0)}(\omega)\right]. \quad (8)$$

We evaluated the MSDs from Eq. (7), using for each Wi the same parameters as in Fig. 3. This leads to the solid lines in Fig. 2, with remarkable agreement. This agreement is even more notable, as we have assumed Eq. (8) to be valid also for finite Wi, so that no more free parameter appears compared to Fig. 3 (the value of $\kappa$ was found to be slightly different for the different Wi, see

Methods section, which we attribute to a small anharmonicity of the potential shown in Fig. 1). Eq. (8) appears well obeyed, so that any notion of effective temperatures is not crucial for understanding the observed results, again in contrast to Ref. [12].

Oscillatory behaviour in physical systems is typically a signature of inertial effects. Indeed, a partial integration in Eq. (1) yields formally an inertial term (see Methods), with an effective mass $10^{10}$ times the actual mass of the particle. More importantly, this mass is negative in equilibrium, whereas it is positive for the parameters used in Fig. 3, e.g., for Wi = 0.04. The observed oscillations may thus formally be attributed to a (positive) particle's mass.

We test this analogy quantitatively, demonstrating that oscillations in an overdamped viscoelastic fluid can be formally described by an underdamped oscillator in an equilibrium bath. This is shown in the inset of Fig. 6 where we plot the experimental MCD for Wi = 0.34, together with the solution of an underdamped oscillator (Methods). Apart from the short-time behaviour, which is fundamentally different for the massive particle, in particular the oscillatory behaviour is very well reproduced (we shifted the theoretical curves in Fig. 6 by an offset time $t_0 = 3.0$ s into negative $t$-direction to map the short-time behaviour correspondingly). The main graph shows the corresponding MSD, which is also in excellent agreement with our experiments. As a result of its inertia, indeed the particle explores a larger phase space at intermediate times. For very large times, the MSD of the massive equilibrium particle approaches $2k_\mathrm{B}T/\kappa$. Thus, the behaviour of a very complex non-equilibrium system appears to be well described by a single—easily experimentally accessible—number, the effective mass. This resembles very much the concept of an effective mass as used for description of conduction electrons[39].

## Discussion

How one can rationalise the presence of particle oscillations within the regime of small Weissenberg number? In contrast to macroscopic rheometric experiments, where a constant rate of shear or stress is imposed, the situation is different when strain or stress is created by a colloidal particle driven by a moving trap. As pointed out by Squires and Brady[7], due to the strong coupling between the colloid and the fluid, the particle's motion is strongly affected by local stress and strain fluctuations. As a result, the rates of both strain and stress become time-dependent, which

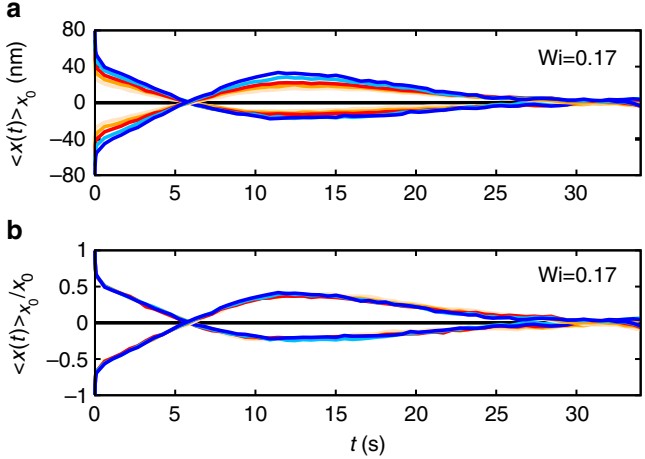

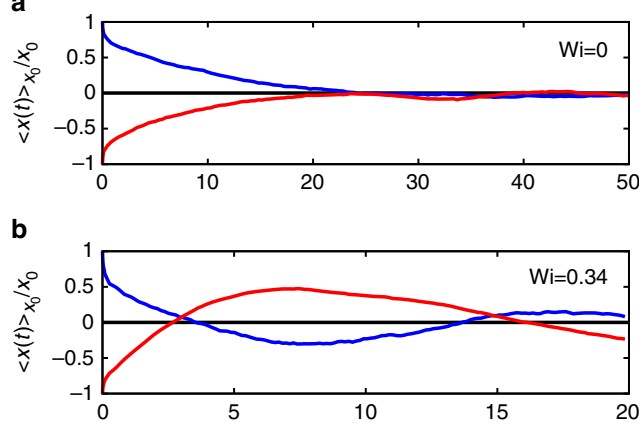

**Fig. 7** Linearity of experimental MCDs. The MCDs are computed for various initial conditions $x_0$ (**a**) (Wi = 0.17). After normalisation on $x_0$ the curves collapse to a single line (**b**)

**Fig. 8** Oscillations in polymer solutions. The MCDs of a particle in a polymer solution are in qualitative agreement with those of the micellar solution, therefore indicating the effect to be generic. **a** Equilibrium, **b** Wi = 0.34

results in an unsteady (oscillatory) particle motion. Such an interpretation is consistent with Fig. 5, where we have shown that oscillations only occur for a certain range of trap stiffnesses. When the trap is too stiff, this corresponds to constant strain rate conditions. When the trap becomes too soft, the particle dynamics is essentially diffusive. In both cases, oscillating modes are excluded.

Our results suggest that underdamped oscillating modes with long correlation times are generally expected for trapped colloidal particles, which are subjected to a non-equilibrium environment with a negative response at long times. Apart from viscoelastic solutions, such conditions should apply to other non-equilibrium baths, e.g., suspensions of active living and synthetic particles, which currently receive considerable attention. In addition, the presence of an underdamped particle dynamics will be of relevance for the use of micrometre-sized colloids as mechanical probes when investigating the dynamics of, e.g., molecular motors or protein complexes within their viscoelastic intracellular environment. The demonstrated continuous variation between underdamped and overdamped modes thereby allows for a large range of tunability.

## Methods

**Preparation of the viscoelastic fluid and the sample**. Our experiments are performed in an equimolar solution of surfactant CPyCl (Sigma-Aldrich, crystalline, 99.0–102.0%) and salt NaSal (Sigma-Aldrich, Reagentplus TM, ≥99.5%) in deionised water at a concentration of 7 mM. After overnight mixing at 318 K, worm-like micelles form and deform dynamically in the solvent[21] with structural relaxation time $\tau_s = 2.5 \pm 0.2$ s, as determined from microrheological measurements at $T = 298$ K[7,19]. The length of such worm-like micelles is between 100 and 1000 nm[22], and the typical mesh size is on the order of 30 nm[23]. Silica microspheres of diameter $2R = 2.73$ μm are highly diluted in this viscoelastic solution. The solution is then inserted into a custom-made chamber with a height of 90 μm. During the measurements, the sample sample cell is thermally coupled to a thermostat at $T = 298 \pm 0.2$ K.

**Particle trapping and tracking**. Optical trapping of a colloidal particle is achieved by a Gaussian laser beam ($\lambda = 1070$ nm), which is tightly focused by a microscope objective (× 100, numerical aperture = 1.4) onto the sample. To avoid hydrodynamic interactions with the walls, the focal plane is adjusted into the middle plane of the sample cell, thus the trap position is more than 40 μm apart from any wall. As confirmed by the particle's displacement distribution, the optical trap corresponds to a harmonic potential $\frac{1}{2}\kappa\xi^2$, where $\kappa$ is the trap stiffness and $\xi$ is the particle position relative to the potential minimum. With a galvanostatically driven mirror, the laser beam and thus the trap position are moved along $\xi$-direction forward and backward over a distance of about 20 μm at constant velocity $v$. The values of $v$ are chosen in a range where the viscosity remains constant, i.e., far away

from shear thinning effects. The smallest velocity that can be achieved with our setup is about 40 nm s$^{-1}$, which yields for our system Wi = 0.04. The centre of mass of the particle is tracked by means of video microscopy at 145 frames per second and a spatial accuracy of 4 nm[40].

**Algorithm of MCD computation from experimental data**. The MCD of a (colloidal) particle is formally defined by $\langle x \rangle_{x_0}(t) \equiv \int dx\, P(x, t|x_0, 0)x$. $P(x, t|x_0, 0)$ is the conditional probability to find the particle at position $x$ at time $t$, given that it was at $x_0$ at time $t = 0$. In the case of discrete experimental data, the MCD at time instant $t_n$ is given by the following weighted sum

$$\langle x \rangle_{x_0}(t_n) = \frac{1}{n(x_0)} \sum_i n(x_i, x_0, t_n) x_i \qquad (9)$$

It is noteworthy that the conditional probability $P(x, t|x_0, 0)$ turned into the corresponding statistical frequency $n(x_i, x_0, t_n)$, i.e., the number of (random) occurrences $x_i$ at time step $t_n$ if the initial position $x_0$ was fixed at $t_0 = 0$. It is normalised by $n(x_0)$, which gives the number of (random) occurrences of equal initial displacements $x(t_n) = x_0$ in a given experimental trajectory. The weighted sum in Eq. (9) runs over all possible outcomes $x_i$ of the experiment.

It is verified that the MCDs are linear in $x_0$ (cf. Figure 7, where we assembled the curves in intervals of $\Delta x_0 = 10$ nm) and therefore we can average over the normalised curves with positive and negative initial condition $x_0$ to improve the overall statistics (see Fig. 3).

**Oscillations in polymer solutions**. The onset of oscillations in the non-equilibrium MCDs is also observed in the case of a semi-dilute polymer solution (polyacrylamide, $M_w = 18 \times 10^6$ at 0.03% wt. in water). In Fig. 8, we show the MCDs for the equilibrium case and Wi = 0.34. The occurrence of oscillations also in such a polymer solution (with a structural relaxation time $\tau_s$ similar to the one of the micellar solution) suggests the effect to be generic for viscoelastic systems with large structural relaxation times.

**Creation of theoretical MCD/MSD curves**. In this subsection, we provide detailed information on the creation of the theoretical MSD and MCD curves in Figs 2 and 3, respectively. As discussed in the main text, the non-equilibrium oscillations in the MCDs are evoked by adding another generic exponential term to the memory kernel with negative amplitude. This approach can be generalised by a sum of exponential functions, i.e.,

$$\Gamma^{(v)}(t) = 2\gamma_\infty \delta(t) + \frac{\gamma_0 - \gamma_\infty - \sum_i \gamma_i}{\tau} e^{-\frac{t}{\tau}} + \sum_i \frac{\gamma_i}{\tau_i} e^{-\frac{t}{\tau_i}}. \qquad (10)$$

It is noteworthy that the time integral of $\Gamma^{(v)}(t)$ equals the zero-frequency coefficient $\gamma_0 \equiv 6\pi\eta R$ for the viscosity $\eta$ at small Weissenberg numbers as determined by the experimental flow curve in Fig. 1c. With this simple model, we adjust the parameters in such a way to obtain best agreement with the experimental MCD and MSD curves. The values of parameters are given in Table 1. It is noteworthy that we allow a slight variation in the trap stiffness $\kappa$. This variation

**Table 1 Values of parameters as used in Figs 2 and 3, respectively**

| Wi | $\kappa$ | $\gamma_\infty$ | $\gamma_0$ | $\tau$ | $\gamma_1$ | $\tau_1$ | $\gamma_2$ | $\tau_2$ |
|---|---|---|---|---|---|---|---|---|
| 0 | 2.8 | 0.18 | 20.6 | 9.1 | — | — | — | — |
| 0.04 | 2.3 | 0.19 | 6.8 | 25.0 | −1135.1 | 27.0 | — | — |
| 0.11 | 2.3 | 0.19 | 5.3 | 28.0 | −204.5 | 17.8 | 109.3 | 10.0 |
| 0.17 | 2.6 | 0.18 | 6.5 | 28.0 | −67.3 | 15.0 | 17.3 | 2.0 |
| 0.24 | 2.7 | 0.21 | 7.1 | 14.8 | −126.4 | 11.0 | 23.0 | 2.1 |
| 0.34 | 2.7 | 0.18 | 6.0 | 16.0 | −84.1 | 12.0 | 6.4 | 0.4 |

The trap stiffness $\kappa$ is given in units of $\mu\mathrm{Nm}^{-1}$, friction coefficients $\gamma$ are given in units of $\mu\mathrm{Nsm}^{-1}$, and memory relaxation times $\tau$ in s

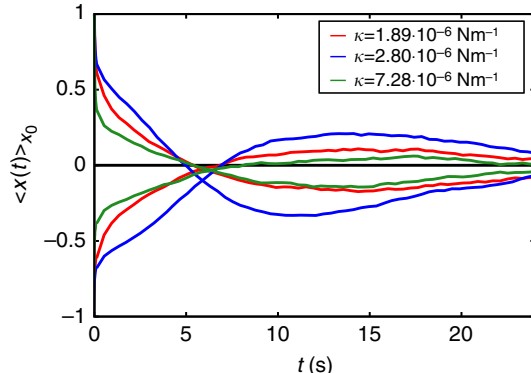

**Fig. 9** Dependence of MCDs on trap stiffness $\kappa$. MCDs plotted for Wi = 0.24 upon varying $\kappa$

incorporates the experimental error in $\kappa$ (e.g., due to a small anharmonicity of the potential, as well as polydispersity between different measurements)

Table 1 reveals that the zero-frequency coefficient $\gamma_0$ experienced by the trapped particle in equilibrium is roughly three times as large as in non-equilibrium. We emphasise that in equilibrium $\gamma_0$ strongly depends on the trap stiffness $\kappa$ and, ultimately, in the limit $\kappa \to 0$, tends to a value comparable to those in non-equilibrium for small Weissenberg numbers (see the value at Wi = 0 in Fig. 1c)), which is obtained from the MSD in the absence of the trap.

It is worth noting that for increasing Weissenberg numbers more exponential terms in Eq. (10) are needed to mimic well the experimental curves.

**Underdamped harmonic oscillator.** The model system of a (stochastic) under-damped harmonic oscillator with mass $m$ quantitatively reproduces the experimental results of the overdamped system as shown in Fig. 6 by adjusting the mass of the particle accordingly (supporting the notion of an effective mass). Here, we give the parameter values, which are used to create Fig. 6. The underdamped equation of motion in the Markovian case reads as

$$m\ddot{x}(t) + \gamma\dot{x}(t) = -\kappa x(t) + f(t). \tag{11}$$

$f(t)$ is delta-correlated Gaussian white noise, i.e., its statistical properties are fully specified by its first two moments $\langle f(t) \rangle = 0$ and $\langle f(t)f(t') \rangle = 2\gamma k_B T \delta(t - t')$. The parameters used for the solid lines in Fig. 6 are $\kappa = 2.8\,\mu\mathrm{Nm}^{-1}$, $m = 32.0$ mg and $\gamma = 5\,\mu\mathrm{Nsm}^{-1}$. It is noteworthy that the value of the friction coefficient $\gamma$ is comparable to the non-equilibrium zero-frequency coefficient $\gamma_0$ in Table 1.

**Absence of oscillations in equilibrium MCDs.** In overdamped dynamics, the Fokker–Planck equation (sometimes also referred to as Smoluchowski equation) is the equation of motion of the probability distribution function (pdf) $P(\Gamma \equiv \{\mathbf{r}_i\})$ of interacting constituents in a system (e.g., colloid and surrounding micelles). It is valid on the Brownian (or diffusive) timescale, where the momentum coordinates of the Brownian particles are relaxed to thermal equilibrium. In this effective description (where the phase space coordinates of the solvent molecules are long relaxed), the time evolution of the pdf is governed by[2]

$$\partial_t P(\Gamma, t) = \Omega P(\Gamma, t)$$
$$\Omega = \sum_{ij} \partial_i \cdot \mathbf{D}_{ij}(\{\mathbf{r}_j\}) \cdot [\partial_j - \beta\mathbf{F}_j]. \tag{12}$$

$\Omega$ is the so-called Fokker–Planck operator containing the $3 \times 3$-dimensional microscopic diffusion matrices $\mathbf{D}_{ij}$ and the total force $\mathbf{F}_j$ acting on particle $j$ ($j = 1 \ldots N$) at position $r_j$. It can be shown that the Hermitian conjugate of the Fokker–Planck operator $\Omega^\dagger = \Sigma_{ij}(\partial_i + \beta F_i) \cdot \mathbf{D}_{ij} \cdot \partial_j$ is Hermitian with respect to the weighted inner product (weighted with the equilibrium pdf $P_{eq}$)[2]

$$\langle g^*\Omega^\dagger h \rangle_{eq} = \langle h\Omega^\dagger g^* \rangle_{eq} = -\left\langle \sum_{i,j} \frac{\partial g^*}{\partial\mathbf{r}_i} \cdot \mathbf{D}_{ij} \cdot \frac{\partial h}{\partial\mathbf{r}_j} \right\rangle_{eq}. \tag{13}$$

Consequently, the eigenvalues $\lambda_n$ of $\Omega^\dagger$ are real and the eigenfunctions $\Omega^\dagger\phi_n = \lambda_n\phi_n$ form a orthogonal basis, i.e., for normalised functions fulfil $\langle \phi_n^*\phi_n \rangle_{eq} = \delta_{nm}$. Moreover, as by definition $\mathbf{D}_{ij}$ is a positive semi-definite matrix[24], we find $\Omega^\dagger$ to be negative semi-definite

$$\langle g^*\Omega^\dagger g \rangle_{eq} \leq 0, \tag{14}$$

i.e. $\lambda_n \leq 0$ for any $n$. All modes are thus strictly overdamped, and any equilibrium correlation function $\langle g(t)g(0) \rangle_{eq}$ can be written as a sum of positive exponentially

decaying functions

$$\langle g(t)g(0) \rangle_{eq} = \left\langle g^*e^{\Omega^\dagger t}g \right\rangle_{eq} = \sum_n |c_n|^2 e^{\lambda_n t}. \tag{15}$$

Specifically, using a linear Langevin equation (cf. Eq. (1)), the MCDs can be directly related to the correlation function of $x$ via

$$\langle x(t) \rangle_{x_0}^{eq} = \beta\kappa x_0 \langle x(t)x(0) \rangle_{eq}, \tag{16}$$

where $\beta = (k_B T)^{-1}$ is the inverse temperature. We conclude that the equilibrium MCDs are strictly monotonic and hence show no oscillatory behaviour for a complex suspension. Another fundamental insight concerning the equilibrium memory kernel $\Gamma^{(0)}(t)$ is obtained by applying the FDT. The FDT relates the linear response function of a system to a small external perturbation to its thermal equilibrium fluctuations. For the trapped Brownian particle we find

$$\Gamma^{(0)}(|t|) = \langle F(t)F(0) \rangle_{eq}. \tag{17}$$

It is worth noting that both sides of the equation implicitly depend on the trap stiffness $\kappa$. Using the same arguments as before, the equilibrium memory kernel is a positive function for all times $t$.

**$\kappa$-dependence of experimental MCDs.** The trap stiffness $\kappa$ appears to be an important parameter for the occurrence of oscillations in the MCDs. In the experiment, $\kappa$ can be varied by changing the intensity of the trap laser. In Fig. 9, we show the normalised MCDs for three different values of the trap stiffness $\kappa$. Apparently, there is a resonant value of $\kappa$ (the one used in the main text) leading to a particularly high oscillation amplitude. For higher and lower $\kappa$, the amplitude decreases, thereby indicating that the resonant behaviour is only present in a certain regime of trap stiffness $\kappa$.

**Mass identification in the Langevin equation.** Oscillations as observed in the MCDs are a feature of inertia. We can corroborate this fundamental principle by reconsidering the generalised Langevin equation in Eq. (1). According to Newton's equation of motion, mass is the proportionality constant in the force/acceleration relation of a massive body. Such a second-order differential equation can be mathematically obtained from Eq. (1) by partial integration. We then find

$$\int_{-\infty}^t ds\,\mathcal{M}^{(v)}(t - s)\ddot{x}(s) = -\gamma_0\dot{x}(t) - \kappa x(t) + f^{(v)}(t). \tag{18}$$

$\gamma_0$ is the friction coefficient at zero frequency, i.e. $\gamma_0 \equiv \int_0^\infty dt\,\Gamma^{(v)}(t)$, and $\mathcal{M}^{(v)}(t - s) \equiv -\int_{-\infty}^s dh\,\Gamma^{(v)}(t - h)$ is identified with the memory kernel of inertia. In this description, the memory of the system is now related to inertial effects while the friction coefficient is time-independent and reduces to the long-time value $\gamma_0$. By mimicking Newton's equation of motion, we may define the mass of the particle as the zero-frequency contribution of $\bar{\mathcal{M}}(\omega)$ and obtain for the memory kernel in Eq. (10),

$$m \equiv \int_0^\infty dt\,\mathcal{M}^{(v)}(t) = -\left(\gamma_0 - \gamma_\infty - \sum_i \gamma_i\right)\tau - \sum_i \gamma_i\tau_i. \tag{19}$$

In equilibrium, the memory kernel $\Gamma^{(0)}$ in Eq. (10) is a sum of positive exponentially decaying functions and therefore $m$ strictly takes a negative value. In

non-equilibrium; however, for the simple model of two exponential functions (cf. Eq. (5)), $m$ may be positive if the amplitude $\gamma_1$ is negative and the relaxation time $\tau_1$ associated with this negative part of the friction kernel is larger than the relaxation time of the positive exponential function. For instance, we find $m = 2.1\,g$ for the parameters used for $Wi = 0.04$ in Table 1.

**Data availability**. The data that support the findings of this study are available from the corresponding author upon reasonable request.

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

## Acknowledgements

We thank Celia Lozano for stimulating scientific discussions and Mike Reppert for discussions during the early stages of this work. C.B. acknowledges financial support from Deutsche Forschungsgemeinschaft (DFG) through the priority programme SPP 1726 on microswimmers and by the ERC Advanced Grant ASCIR (Grant No. 693683). J. R.G.-S. was supported by DFG Grant Number GO 2797/1-1. M.K. and B.M. were supported by DFG Grant Number KR 3844/3-1, and M.K. through DFG Grant Number KR 3844/2-1.

## Author contributions

J.B. performed the experiments. B.M. performed the numerical and theoretical calculations. The data analysis and the manuscript preparation was performed by all authors.

## Additional information

**Competing interests:** The authors declare no competing interests.

