## [Peer Review File · Nature Communications]

Reviewer #1 (Remarks to the Author):

Dear authors,

I have completed my review of your manuscript.

My principal concerns with the manuscript center around the transition between the over damped and oscillatory response. In the manuscript it is presented as if there is no Weissenberg number below which the overdamped response should emerge. Why? Figure 4 describes dynamical phase boundaries in terms of the proposed model parameters. The experiments are represented with a red dot, but the axes are scaled with respect to arbitrary units. How can this be? What are the molecular origins of the negative memory modes in the rheology of the micellar solutions? The idea that the loose trapping regime could produce strange responses in nonlinear microrheology dates back to the original theoretical work in the area by Squires and Brady. This is virtually uncredited in the manuscript.

I found the work interesting but not suitable for Nature Communications.

Reviewer #2 (Remarks to the Author):

In this manuscript the motion of a colloidal particle dragged through a viscoelastic medium is studied. Though the motion occurs in the Stokes regime at small driving velocities, an oscillatory response is found. The behavior is interpreted and described by a negative friction memory kernel and a simple model of an underdamped oscillator is found to describe the main trends correctly.

This papers reports on an important and fundamental problem, namely colloidal motion of non-Newtonian solvents. The behavior found is clearly described within a simple intuitive picture. The comparison between experiment and theory is insightful.

I think that this is a good case for a publication in Nature Communications. But before publication the authors should consider the following points.

1) How specific is the worm-like micellar solution used as a viscoelastic solvent? It is claimed on page 5 that the results will hold for any non-equilibrium environment with a negative response at long times? Is there any other example where the same behavior has been seen? The authors have studied related systems (in [14], [16] and in the PRL 2016 with Blokhuis) in other viscoelastic solvents (polymer mixtures, entangled lambda-phage DNA). What about the response of these or other solvents at low drive?

2) This continues on a similar line. The phenomenological interpretation is in terms of a negative memory. But why comes this about microscopically and how can this be linked to the dynamics of the wormlike micellar solvent? Are interactions between the colloidal surface and the micellar solvent important (repulsive, attractive etc) or is it a pure bulk property of the micellar solution? What is the typical largest size of the micellar network structure? This should be much smaller than the colloidal radius. And what are the nonequilibrium excitations in the micellar network which give rise to the oscillations: damped phonons,

topological excitations or anything else? Explaining these points would be highly beneficial for the reader.

3) On page 1 τ_s is called a structural relaxation time. Is this the time on which the topology and connectivity of the entangled network breaks down or does it rather measure the fluctuations of the topologically fixed network? Again, this is important for the fundamental microscopic interpretation, see 2).

4) Figure 2: It is stated that the MSD in equilibrium are monotonic but it is not explicitly stated that the MSDs under higher Wi are non-monotonic. So, are the MSDs nonmonotonic for any nonzero Wi ? The plots in Figure 2 and the fits from Eq. (7) seem to support a non-monotonic behavior. Can the fits be extended towards a longer time to see 1-2 oscillations towards the asymptotics? This would a lot I think.

5) Minor: The authors provide the Weissenberg number Wi , they should also provide the Reynolds number Re or state that Re is very small.

6) Minor: 2 References on page 1 are not given clearly but as a question mark (at least in my printout of the pdf-file provided). This should be corrected.

Dear Editor,

We are grateful for forwarding the reports concerning our work on "Oscillating Modes of Driven Colloids in Overdamped Systems". Referee 2 states "This papers reports on an important and fundamental problem" and "I think that this is a good case for a publication in Nature Communications". Referee 1 raised some issues, which have been carefully addressed and clarified in the revised manuscript. In the resubmitted version, we addressed the comments of both referees point by point and hope that the revised manuscript is now suitable for publication in Nature Communications. To facilitate tracking of changes in the revised manuscript, we have marked them in blue in the resubmitted version.

Reviewer 1

1. *My principal concerns with the manuscript center around the transition between the overdamped and oscillatory response. In the manuscript it is presented as if there is no Weissenberg number below which the overdamped response should emerge. Why?*

It was not our intention to suggest that there is a discontinuous transition between the overdamped and the underdamped regime. In fact, we expect a gradual change from the underdamped to the overdamped case. Perhaps the referee was misled by Fig.3 where we demonstrated particle oscillations for several Wi down to $Wi=0.04$ and the absence of oscillations for $Wi=0$.

It should be mentioned that experimentally, there exists a limit in reaching arbitrarily small Weissenberg numbers. In our setup, the lowest possible value of $Wi=0.04$ is determined by the smallest experimentally achievable translational velocity (about 40nm/sec) of the optical trap (this information is now also added to the main text and the Methods). In addition to such experimental limitations, however, the detection of oscillations becomes increasingly difficult as Wi decreases. This is, because the oscillation amplitude decreases when decreasing Wi (see Fig.3). In addition, the oscillation period increases with decreasing Wi . The combination of both effects makes the detection of particle oscillations increasingly difficult with decreasing Wi , in particular when considering the presence of Brownian fluctuations.

To make this point more clear, we have added as a new Fig.4 the dependence of the oscillation amplitude and the oscillation period vs. Wi . From this dependence, the rather smooth decrease of both quantities is seen which suggests a continuous disappearance of the oscillations when approaching $Wi=0$.

Changes:

New Fig.4 with the following caption:

'Oscillation amplitude and frequency. Oscillation amplitude (left) and frequency (right) of the MCD curves (shown in Fig. 3) vs. Wi . The insets illustrate, how these quantities are derived from the experimental data. Both curves decrease with decreasing Weissenberg number, i.e., towards equilibrium. Regarding frequency, this implies that oscillations are

particularly slow at small Wi . Since the second root is not visible for $Wi=0.04$ in Fig. 3, the corresponding value of Ω has been obtained from the time difference between the extremum and the first root.'

Added the statement on page 3, left column, that $Wi=0.04$ is the slowest accessible driving velocity.

Additional sentence on page 3:

'Fig. 4 shows the dependence of amplitude and frequency of oscillations on Weissenberg number, were for both quantities, a gradual decrease towards equilibrium ($Wi=0$) is observed.'

Additional sentence in Methods B:

'The smallest velocity which could be achieved with our setup was about 40nm/s which yields for our system $Wi=0.04$.'

2. *Figure 4 describes dynamical phase boundaries in terms of the proposed model parameters. The experiments are represented with a red dot, but the axes are scaled with respect to arbitrary units. How can this be?*

We fully agree, the use of a.u. was misleading. In fact, the axes have been normalized with respect to the values which were used for fitting to the experimental curves, so that the red dot is placed exactly at (1,1). In the new version of Fig.4 (now Fig.5), we have made this now explicitly clear by relabelling the axes " $\kappa/\kappa_{\text{exp}}$ " and " $\gamma_1/\gamma_{1,\text{exp}}$ ", while the graph remains the same otherwise. We have also adjusted the figure caption, which now states explicitly how the axes are normalized.

Changes:

Changed caption of Fig.5 (formerly Fig.4):

Dynamical phase diagram. The model of Eq. (5) yields distinct solutions, either decaying exponentially or showing oscillations, here shown for $Wi=0.04$ as a function of κ and γ_1 normalised on the corresponding experimental parameters $\kappa_{\text{exp}} = 2.3 \mu\text{N/m}$ and $\gamma_{1,\text{exp}} = -1135.1 \mu\text{Ns/m}$, i.e., those obtained from the fit curve in Fig.3.'

3. *What are the molecular origins of the negative memory modes in the rheology of the micellar solutions?*

Regarding the molecular origin, we have a similar picture in mind as usually invoked to explain the behaviour of start-up curves in rheological measurements of complex systems, which has been found relevant for glassy materials (see for example Ref. [20] of our manuscript or the following references *Journal of Rheology* **57**, 149 (2013), *Phys. Rev. E* **58**, 738 (1998), *Journal of Rheology* **44**, 323 (2000), *Annu. Rev. Condens. Matter Phys.* **2**, 353 (2011).)

When starting to shear a (dense or glassy) suspension, the measured stress grows as a function of time, and can then go through a maximum (the yield stress), a phenomenon called the stress overshoot. Such overshoot is mathematically described by a (stress) memory function, the shear modulus, which is negative for long times and thus reminiscent to the memory functions used in our manuscript. Although our experiments are in a steady driving mode, the particle position fluctuates in the trap. A repeated cycle-scenario of building up stress until the micellar network yields could thus be the origin of the observed effects. A microscopic theory demonstrating the negative parts of the shear modulus is given in Ref. [20] and *Journal of Rheology* **57**, 149 (2013).

To emphasize, that the idea of negative friction terms have been previously discussed in other systems, we have added the above mentioned references to the revised version of the manuscript.

Changes:

Added paragraph on page 3f.:

'Negative memory is a concept which has been used in other fields of rheology of complex systems, for example when applying macroscopic shear. If such shear is started abruptly, one sometimes observes so called stress overshoots, where the stress goes through a maximum as a function of time, once the yield stress is overcome [20, 32–35]. Theoretically, these overshoots have been described by negative memory, as found from microscopic derivations in Refs. [20,35].'

Included sentence on page 4:

'Conceptually different to the above mentioned studies on macroscopic shear, it is the additional presence of the optical trap, which, in combination with the negative memory yields oscillatory solutions in our experiments.'

4. *The idea that the loose trapping regime could produce strange responses in nonlinear microrheology dates back to the original theoretical work in the area by Squires and Brady. This is virtually uncredited in the manuscript.*

We fully agree, that the work by Squires and Brady is of central importance in nonlinear microrheology, therefore we had already cited it as Ref. 7. This paper addressed for the first time, that there is a strong dependence of micro-rheological observations on the trap stiffness. Depending on the strength of the trap, a driven colloidal particle can behave as a constant force or constant velocity probe or a mixture of these modes. Also, the authors emphasise the general importance of non-equilibrium micro-structural deformations on the motion of a forced probe particle which is eventually the origin of the particle oscillations observed by us (even though, such oscillations were not mentioned in this work).

In the second paragraph of the revised version of the manuscript, we refer now specifically to the trap stiffness dependence in the context of rheological measurements as discussed in the work of Squires and Brady. In addition, we now refer to their work in the second last paragraph when discussing, that in contrast to macroscopic experiments where a constant

rate of shear or stress is imposed externally, this may no longer be true when the particle is driven by an optical trap.

Changes:

Page 1:

'Theoretical studies predicted, that in this regime the particle dynamics becomes largely affected by the fluid's non-equilibrium microstructural deformations and that the measured viscosity may exhibit a non-trivial dependence on the trap stiffness (Ref. [7]).'

Page 6:

'As pointed out by Squires and Brady [7], due to the strong coupling between the colloid and the fluid, the particle's motion is strongly affected by local stress and strain fluctuations.'

Reviewer 2

In this manuscript the motion of a colloidal particle dragged through a viscoelastic medium is studied. Though the motion occurs in the Stokes regime at small driving velocities, an oscillatory response is found. The behavior is interpreted and described by a negative friction memory kernel and a simple model of an underdamped oscillator is found to describe the main trends correctly.

This papers reports on an important and fundamental problem, namely colloidal motion of non-Newtonian solvents. The behavior found is clearly described within a simple intuitive picture. The comparison between experiment and theory is insightful.

1. *How specific is the worm-like micellar solution used as a viscoelastic solvent? It is claimed on page 5 that the results will hold for any non-equilibrium environment with a negative response at long times? Is there any other example where the same behavior has been seen? The authors have studied related systems (in [14], [16] and in the PRL 2016 with Blokhuis) in other viscoelastic solvents (polymer mixtures, entangled lambda-phage DNA). What about the response of these or other solvents at low drive?*

We are particularly thankful for this question. As mentioned in the third paragraph and in the Methods D section (Fig. 8), we observed the oscillations also in a semi-dilute polymer polyacrylamide solution (this also applies to the W_i -dependent mean-square displacement) which is a visco-elastic fluid with a similar relaxation time as the the worm-like micellar solution. It should be mentioned, that the chemistry and the structure of both systems are rather different, therefore we assume, that the observed oscillations are a generic effect of viscoelastic solvents.

After reading the referee's question, we felt, that this point was not highlighted enough in the paper. Therefore, in the revised version of the manuscript, we have emphasized, that our findings are not restricted to a single viscoelastic fluid and that we believe that this is a generic effect.

Changes:

Added paragraph on page 1:

'While the main text focuses on a worm-like micellar solution, we observe similar particle oscillations in other viscoelastic fluids comprising different chemistry and microstructure (see Methods 1 D). Therefore, we believe that the reported oscillations are a generic feature of particles in non-equilibrium baths.'

2a. This continues on a similar line. The phenomenological interpretation is in terms of a negative memory. But why comes this about microscopically and how can this be linked to the dynamics of the wormlike micellar solvent?

Negative memory is a concept which has already been used or observed in other fields of rheology of complex systems, for example when applying macroscopic shear. If such shear is started abruptly, one sometimes observes so called stress overshoots, where the stress goes through a maximum as a function of time, once the yield stress is overcome. Theoretically, these overshoots have been described by negative memory. Ref. [20] provides a microscopic derivation of such negative memory, however, for the considered case of microrheology, a microscopic description is still lacking. Conceptually different to the above mentioned studies on macroscopic shear, it is the additional presence of a restoring force due to the optical trap, which, in combination with the negative memory yields oscillatory solutions in our experiments. The excited modes of the micellar network are thus those responsible for the negative memory.

Changes:

Added paragraph on page 3f.:

'Negative memory is a concept which has been used in other fields of rheology of complex systems, for example when applying macroscopic shear. If such shear is started abruptly, one sometimes observes so called stress overshoots, where the stress goes through a maximum as a function of time, once the yield stress is overcome [20, 32–35]. Theoretically, these overshoots have been described by negative memory, as found from microscopic derivations in Refs. [20,35].'

Included sentence on page 4:

'Conceptually different to the above mentioned studies on macroscopic shear, it is the additional presence of the optical trap, which, in combination with the negative memory yields oscillatory solutions in our experiments.'

2b. Are interactions between the colloidal surface and the micellar solvent important (repulsive, attractive etc) or is it a pure bulk property of the micellar solution? What is the typical largest size of the micellar network structure? This should be much smaller than the colloidal radius. And what are the nonequilibrium excitations in the

micellar network which give rise to the oscillations: damped phonons, topological excitations or anything else? Explaining these points would be highly beneficial for the reader.”

This is an interesting point. As already mentioned, similar particle oscillations have been also observed in chemically and structurally rather different systems with comparable relaxation times. Therefore, specific interactions between the fluid and the particle are unlikely to be the origin of the oscillations. We did not find indications of attractive interactions of the micellar solution and the colloidal particle (e.g. an adsorption layer of micelles) because the translational diffusion coefficient ($W_i=0$) was in agreement with the theoretical value obtained from the geometrical particle size. Repulsive interactions should be short-ranged since the Debye screening length is on the order of few tens of nanometers. The largest size of the micellar network structure is the mesh size which is about 30 nm (see *Phys. Rev. E* **72**, 011504 (2005)), and thus much smaller than the particle size. Contrary to polymer networks, micellar structures form highly dynamic networks. In addition to reptation known from polymer networks, the micellar chains break and reform permanently. Apart from this, we have not found further information about the dynamics of our system in the literature.

Changes:

Added to Page 2:

‘[...]’, where the length of the worm-like micelles is typically found in between 100 - 1000 nm [22] and the typical mesh size is on the order of 30 nm [23]’

Added to Methods A:

‘[...]’and the typical mesh size is on the order of 30 nm [23].’

3. *On page 1 τ_s is called a structural relaxation time. Is this the time on which the topology and connectivity of the entangled network breaks down or does it rather measure the fluctuations of the topologically fixed network? Again, this is important for the fundamental microscopic interpretation, see 2).*

Before discussing the meaning of the relaxation time τ_s , let us briefly recap, how it is measured: τ_s is determined from a recoil experiment, where the probe particle is dragged at constant velocity through the micellar solution thus inducing strain. When the trapping force is suddenly removed, the particle recoils until the complete microstructural recovery of the fluid (see Ref. 19). From the recoil dynamics, we obtain the structural relaxation time of the fluid.

For wormlike micelles, the structural stress-relaxation time τ_s results from a combination of reptation, breakage and reformation dynamics. The basic concept is that the stress associated with a segment of a wormlike micellar tube is relaxed when a chain end has had sufficient time to pass through it. Because the chain end has a finite lifetime, it is able to reptate a curvilinear distance along its tube over a time τ_{rep} . In addition to reptation

providing a pathway for stress relaxation, an end can form due to chain breakage. Provided that a chain end is formed by a break during a time τ_{break} within the same curvilinear distance from that tube segment, then that chain end may pass through the segment, relaxing the stress. The time taken for this to occur is $\tau \sim \sqrt{\tau_{\text{rep}} \times \tau_{\text{break}}}$. Stress relaxation via this mechanism involves all tube segments equally. In the so-called 'fast-breaking limit' where $\tau \gg \tau_{\text{break}}$, any given chain undergoes many breakage and recombination events before sufficient time elapses for it to reptate through its tube. Memory of the initial length or configuration of the chain is therefore erased and all tube segments are described by the same relaxation time.

Changes:

Bottom of page 1: with a structural relaxation time $\tau_s = 2.5 \pm 0.2$ s determined by a recoil experiment [19]

4. *Figure 2: It is stated that the MSD in equilibrium are monotonic but it is not explicitly stated that the MSDs under higher Wi are non-monotonic. So, are the MSDs nonmonotonic for any nonzero Wi ? The plots in Figure 2 and the fits from Eq. (7) seem to support a non-monotonic behavior. Can the fits be extended towards a longer time to see 1-2 oscillations towards the asymptotics? This would a lot I think.*

The experimental MSD curves indeed suggest a non-monotonic behaviour for non-zero Weissenberg numbers. In this respect, the MSD curves bear a resemblance to the MCD curves which also show this non-monotonicity in non-equilibrium. In support of this notion, we decided to extend the fit curves to the same timescales as the corresponding MCD curves. This can be seen in the changed Fig. 2.

Changes:

Fig. 2:

Extended the fit curves to the same timescales as the corresponding MCD curves.

5. *Minor: The authors provide the Weissenberg number Wi , they should also provide the Reynolds number Re or state that Re is very small.*

We estimate the Reynolds number to be $Re=0.3e-9$ at Weissenberg number of $Wi=0.04$, so that inertial effects in the fluid are negligible. We agree with the referee that this is an important point.

Changes:

Page 2:

We added the estimate of the Reynolds number on page 2, left column.

6. *Minor: 2 References on page 1 are not given clearly but as a question mark (at least in my printout of the pdf-file provided). This should be corrected.*

We apologise for the inconvenience.

Changes:

Correction of the broken references in the resubmitted version.

Additional minor changes:

The MSD and MCD curves in Fig. 2, 3 and 6, respectively, have been slightly changed due to a careful double checking of the computation. The adjusted parameters are indicated in blue color in footnote 1 on page 5, as well as in Table I and Methods F. on page 9.

The molecular weight of polyacrylamide given in Methods D. has been corrected, now stating the correct number $M_w = 18e6$ at 0.03% wt. in water.

Yours sincerely,

Clemens Bechinger
on behalf of all the authors

Reviewer #1 (Remarks to the Author):

The authors have answered all my critiques. Thank you.

Reviewer #2 (Remarks to the Author):

The authors have revised the manuscript taking all my concerns into account and I recommend publication in Nature Communications in present form.